# Adnexal Masses in Pregnancy: A Single-Centre Prospective Observational Cohort Study

**DOI:** 10.3390/diagnostics14192182

**Published:** 2024-09-30

**Authors:** Jonathan Gaughran, Catherine Magee, Sian Mitchell, Caroline L. Knight, Ahmad Sayasneh

**Affiliations:** 1Women’s Health, Guy’s and St Thomas’ Hospitals NHS Trust, London SE1 7EH, UK; jonathan.gaughran@gstt.nhs.uk (J.G.); catherine.magee4@nhs.net (C.M.); sian.mitchell@gstt.nhs.uk (S.M.); 2School of Life Sciences, King’s College London, London SE1 7EH, UK; caroline.knight1@gstt.nhs.uk; 3Fetal Medicine, Guy’s and St Thomas’ Hospitals NHS Trust, London SE1 7EH, UK; 4Gynaecological Oncology, Guy’s and St Thomas’ Hospitals NHS Trust, London SE1 7EH, UK

**Keywords:** adnexa uteri, cyst, ovary, pregnancy, screening, ultrasound

## Abstract

Objective: To prospectively determine the nature of adnexal masses diagnosed during pregnancy and investigate whether ultrasound was a reliable means of assessing these. Methods: A single-centre prospective observational cohort study was conducted in a large tertiary referral hospital in London. Pregnant women with an adnexal mass detected at or prior to the 12-week routine ultrasound received a detailed ultrasound by a level II ultrasound practitioner at the time of detection; at 12 weeks; 20 weeks; and 6 weeks postpartum. The following outcomes were recorded: subjective impression of the mass; International Ovarian Tumor Analysis simple rules classification; resolution and intervention rate; and the incidence of complications related to the mass. Results: A total of 28,683 pregnant patients were scanned and an adnexal masses was detected in 277 patients, yielding an incidence of 1%. 274 participants were included in the analysis. Subjective impression was as follows: simple 75.9%; dermoid 29.1%; endometrioma 6.6%; haemorrhagic 3.3%; para-ovarian 2.6%; torted simple 0.7%; decidualized endometrioma 0.4%; fibroma 0.4%; theca luteal 0.4%; and borderline ovarian tumour: 0.7%. There was a significant reduction in the volume at each scan (*p* < 0.0001). Approximately 74.2% of the masses resolved spontaneously. Surgery was performed in 14/274: 2 antenatally, 6 at caesarean section, and 6 postpartum. In 5/247 (2%), there were complications due to the mass. Using IOTA simple rules, 272/274 (99.3%) (*p* < 0.0001) were classifiable. Only 1/274 (0.4%) had malignant features as per IOTA (*p* = 0.05). As there were no confirmed malignancies, the accuracy of IOTA simple rules could not be calculated. Conclusions: Adnexal masses in pregnancy are uncommon and the majority spontaneously resolve. Malignancy is rare, as are complications. In the absence of concerns regarding malignancy or cyst accident, there is no need for additional monitoring of these masses during pregnancy.

## 1. Introduction

Adnexal masses in pregnancy are uncommon, with the incidence ranging from 1 in 76 to 1 in 2328 [1]. Their incidence appears to be rising which is likely to be multifactorial [2]. As ultrasound technology and its availability improves, detection is likely to increase [3]. Additionally, due a steady rise in the age at which women are having their first child, matched with the fact that both benign and malignant adnexal masses are more common with advancing age, the prevalence of adnexal masses in pregnancy is likely to increase [4,5].

The majority are asymptomatic and detected during routine antenatal care [6,7]. Malignancy in this cohort is rare, and the risk of adnexal torsion is 1–6% lower than in non-pregnant women [8,9,10]. As operating during pregnancy is associated with a risk of adverse foetal and maternal outcomes, conservative management is favoured when safe [11].

The assessment of the adnexa during routine antenatal ultrasound is a form of opportunistic screening, for which there is no evidence basis. Ultrasound may be less reliable in pregnancy for many reasons. As the uterus expands, the adnexa may not be visible transvaginal, meaning the transabdominal approach must be employed which is believed to be less accurate [12]. Additionally, alterations in utero-ovarian blood flow may alter Doppler findings [13]. Endometriomas may undergo alterations driven by hormonal changes known as decidualization which mimic malignancy on ultrasound, and rarer masses specific to pregnancy such as luteomas may be detected [14,15,16].

While guidelines exist on the management of adnexal mass in pre- and post-menopausal women, currently none exist for pregnancy, and ultrasound tools such as International Ovarian Tumor Analysis (IOTA) have not been validated in pregnancy [4,17]. With a poor evidence base and a potentially growing incidence, the need to offer evidence-based guidance is paramount [18].

The primary aim of this study was to determine the nature of adnexal masses diagnosed in pregnancy and to monitor their spontaneous resolution, complication, and intervention rate. The secondary aims were to determine if ultrasound by a level II practitioner is a reliable means of assessing this pathology, to assess whether IOTA simple rules are accurate in pregnancy, and when available, to compare histology to ultrasound findings.

## 2. Materials and Methods

This was a single-centre prospective observational cohort study conducted from January 2019 to August 2021, with approval from the South London Research & Ethics Committee (18/LO/1033). Potential participants were identified either while attending the Early Pregnancy Unit (EPU) for an emergency ultrasound scan in early pregnancy, or by staff performing routine antenatal dating scans. Any pregnant woman (confirmed on ultrasound or by positive urinary or serum human chorionic gonadotropin) over the age of 16 years with at least one adnexal mass (excluding corpora lutea of <30 mm) was eligible [7]. Anyone with a personal history of ovarian malignancy or a Borderline Ovarian Tumor (BOT) was excluded. Patient demographics and medical history were recorded. All participants received study-specific ultrasound assessments of their adnexal mass at the time of detection, at the time of the routine dating scan (11–14 weeks), at the anomaly scan (18–22 weeks), and approximately 6 weeks postpartum. The timing of the ultrasound scans was selected to correspond with critical points in pregnancy management: detection or routine dating scan (around 12 weeks) allowed for early identification of adnexal masses; the anomaly scan (18–22 weeks) enabled further assessment as the pregnancy progresses and the uterus expands, which may impact visibility and mass characteristics; and the postpartum scan at 6 weeks was included to assess the natural resolution of masses after pregnancy when most adnexal masses tend to resolve spontaneously. Scans were performed by level II ultrasound practitioners with appropriate expertise and certification in gynaecology ultrasound [19,20]. All images were subsequently reviewed by level III ultrasound practitioners from the research group (JG or AS), if disagreements or uncertainty occurred, the patient was asked to return for a further scan by a level III practitioner. All ultrasound scans were performed using a Voluson ^®^ E8 or E10 (GE Healthcare Ultrasound, Milwaukee, WI, USA) and findings were documented on Astraia© (Ismaning, Germany). Early pregnancy, 12-week, and postpartum scans were performed transvaginal as the default with conversion to transabdominal as required, using the machine’s ‘Gynecology’ pre-setting. The default for all 20-week scans was a transabdominal approach due to displacement of the adnexa by the expanded uterus and the machine’s ‘obstetric first trimester’ pre-setting was used. The adnexal mass volume (mL) was calculated using the prolate ellipsoid formula (L × H × W × 0.52). A subjective impression based on pattern recognition was assigned and IOTA simple rules features were recorded using a pre-populated proforma on Astraia©, and the resultant impression of ‘benign’, ‘malignant’, or ‘unclassifiable’ was documented [21]. The hospital’s electronic records were accessed for operative notes as well as MRI and histology results. According the consent form, any incomplete data from patients who withdrew or were lost to follow-up were included in the analysis.

Statistical analysis was performed using MedCalc® (MedCalc version 20.010, Ostend, Belgium, 2018) [22]. A power calculation was based on the likelihood of the diagnosis of the least common complication of adnexal masses in pre-menopausal women: malignancy and was determined to be a sample size of 26,616. Statistical significance was determined as a *p*-value of <0.05. A 95% confidence interval was calculated for incidence rates, paired t-tests were used to assess changes in the adnexal mass volume, and a two-way Chi-squared test was employed to assess the correlation between variables [23,24].

## 3. Results

During this 31-month period, 13,956 pregnant patients were scanned in the EPU as an emergency, and 14,727 routine 12-week antenatal scans were performed giving a total of 28,683. Adnexal masses were detected in 277 patients, yielding an incidence of 1% (277/28,683). Two patients declined to participate in the study and one was not eligible due to a recent diagnosis of a serous borderline ovarian tumour BOT. As such, 274 patients were included in the analysis. The mean age at diagnosis was 32.03 years (range: 19–45 years). Ethnicity was documented as the following: White in 165/274 (60.2%); Black in 80/274 (29.2%); Asian in 22/274 (8.0%); Arabic in 6/274 (2.2%) and Mixed Race in 1/274 (0.4%). The mean gravida and parity were 1.94 (median: 1.00) and 0.53 (median: 0.00), respectively. The pregnancy was conceived using Assisted Reproductive Techniques (ART) in 15/274 (5.5%) participants. Pregnancy loss (6 miscarriages and 1 termination) occurred in 7/274 (2.6%; incidence rate: [95% CI: 0.01–0.05]) participants, all of which were prior to 12 weekgs gestation—this data was included in the analysis. Symptoms of lower abdominal/pelvic pain or discomfort were reported in 24/274 (8.8%); incidence rate 0.09 (95% CI 0.06–0.13).

Of the 274 women who participated in this study, a unilateral mass was detected in 266/274 (97%) and a bilateral mass in 8/274 (3%). The total adnexal mass count was 282. A total of 21/274 (7.7%; incidence rate 0.77 [95% CI 0.05–0.12]) participants were lost to follow-up, all of whom had a single mass.

An adnexal mass was detected in 114/274 (41.6%) participants during an emergency presentation to the EPU prior to 12 weeks gestation. In 24/114 (21.1%) of these participants, abdominal pain was the reason for presentation, while in 90/114 (78.9%) it was due to vaginal bleeding or hyperemesis. By the time of the 12-week scan, in 17/114 (14.9%) participants, the mass had spontaneously resolved.

At the routine 12-week scan, a mass was detected in 250/274 (91.2%) of participants. These were new masses, detected in asymptomatic participants in 153/250 (61%), while in 97/250 (39%) participants, this was a persistent mass detected during an emergency scan earlier in their pregnancy. By the time of the 20-week scan, in 73/250 (29.2%) participants, the mass had resolved, and 7 participants were lost to follow-up, reducing the denominator to 243.

At the 20-week scan, a mass was detected in 170/243 (70%) of the participants, all of which were believed to be persistent masses detected during the 12-week scan. By the 6-week postpartum scan, in 104/170 (61%) participants, the mass had resolved and 14 were lost to follow-up, further decreasing the denominator to 156.

At the 6-week postpartum scan, a mass was detected in 66/156 (42%) participants. None of these were believed to be new masses, but rather persistent masses detected at the 20-week scan. Overall, in 74% of the participants, the mass spontaneously resolved by the postpartum scan (Figure 1).

The mean volume of the adnexal masses decreased significantly throughout the study. Prior to the 12 weeks, the mean volume was 47.24 cm^3^ (95% CI: 44.09–50.39 cm^3^). At the 12-week scan, the mean volume was 32.09 cm^3^ (95% CI: 28.29–35.88 cm^3^); a reduction of −15.15 cm^3^ (95% CI: −18.47 to −11.83; *p* < 0.0001). At the 20-week scan the mean volume was 22.82 (95% CI: 19.96–26.69 cm^3^); a reduction of −15.56 cm^3^ (95% CI: −17.79 to −13.33; *p* < 0.0001). At the 6-week postpartum scan, the mean volume was 9.94 cm^3^ (95% CI: 7.68–12.20 cm^3^); a further reduction of −12.36 cm^3^ (95% CI: −14.89 to −9.84; *p* = 0.0001). (Figure 2).

Subjective impression was as follows: simple 208/274 (75.9%); dermoid 25/274 (9.1%); endometrioma 18/274 (6.6%); haemorrhagic 9/274 (3.3%); para-ovarian 7/274 (2.6%); torted simple 2/274 (0.7%); decidualized endometrioma 1/274 (0.4%); fibroma 1/274 (0.4%); theca luteal 1/274 (0.4%); and mucinous BOT 2/274 (0.7%). In the eight patients who had bilateral masses, the subjective impression was the same for both sides. In the 14 cases where histology was available, the subjective impression was correct in 11/14 (79%) and incorrect in 3/14 (21%). During follow-up, there was a change in subjective impression in 14/274 (5.1%) which was not statistically significant (Chi-squared test *p* = 0.31). The changes were as follows: simple changed to endometrioma (4); simple changed to haemorrhagic (3); haemorrhagic changed to simple (3); endometrioma to haemorrhagic (1); simple to dermoid (1) ovarian changed to para ovarian (1); and para ovarian changed to ovarian (1).

There was a statistically significant correlation between the subjective impression and resolution rate (Chi-Squared test *p* < 0.0001), with simple cysts being the most common to resolve (Table 1).

A level III gynaecology ultrasound practitioner was asked to review and confirm all the aforementioned cases in which there was a change in subjective impression, the two suspected BOT and one other case. As such, their input was deemed necessary in 18/274 (6.6%) of patients. In all cases, the level III practitioner agreed with the level II practitioner, except for one where the subjective impression was changed from endometrioma to fibroma.

MRI was used in 1/274 (0.4%) participants. Due to this low number, an agreement rate between MRI and ultrasound, and MRI and histology could not be calculated. In this one case, both ultrasound and MRI gave an impression of mucinous borderline ovarian tumour BOT. The subsequent histological diagnosis was a dermoid cyst.

Surgery to remove the adnexal mass was performed in 14/274 (5.1%) participants. There was no statistically significant difference in the timing of surgery (Chi-squared test, *p* = 0.32): antenatally in 2/14 (14.3%), at the time of caesarean section in 6/14 (42.9%), and postpartum in 6/14 (42.9%). Of the two surgeries performed in the antenatal period, one was a confirmed torsion of a simple cyst and the second was a case with a subjective impression of mucinous BOT but a histological diagnosis of a dermoid cyst. There was no malignancy or BOT diagnosed in the surgical group. A summary of the histological diagnosis is presented in Table 2.

After excluding the seven participants who experienced first-trimester pregnancy losses and the twenty-one participants lost to follow-up, complications due to the adnexal mass occurred in 5/246 (2%) participants. In one case, a simple cyst ruptured and bled leading to significant hemoperitoneum, subsequent intra-abdominal infection, and premature delivery of a live infant at 26 weeks gestation. The participants underwent antenatal surgery as described above, and another two presented with suspected torsion of simple cysts, underwent transabdominal drainage with resolution of symptoms, and had an uncomplicated pregnancy thereafter.

Using IOTA simple rules, 272/274 (99.3%) were classifiable, and 2/274 (0.7% were unclassifiable (*p* < 0.0001). Only 1/274 (0.4%) had malignant features (*p* = 0.05). The latter ovarian cyst was a multilocular cyst with increased vascularity (IOTA colour score of 4). The patient had the ovarian cyst removed after delivery and it was a mature cystic teratoma. As there was no malignancy in this cohort, the diagnostic performance of IOTA simple rules could not be calculated.

## 4. Discussion

This study shows that adnexal masses in pregnancy are uncommon, the majority are incidental findings that will spontaneously resolve; complications are uncommon and malignancy is rare. While the accuracy of IOTA in pregnancy is yet to be established, this study suggests it is likely to be applicable.

There are a number of strengths to this study. At the time of writing, this was the largest prospective study assessing adnexal masses in pregnancy and the second to assess IOTA simple rules in this cohort [25]. The inclusion of women of all ages and ethnic backgrounds as well as those who had conceived as a result of ART should increase the generalizability. The fact that all scans were performed by level II ultrasound practitioners from medical, nursing, and sonography backgrounds, as part of emergency and routine antenatal care, should further increase the generalizability of this paper as it reflects the geographical variations in practitioners performing these scans [26]. Each patient was not followed up by a single operator, which is again reflective of standard practice and may reduce bias. As evidence suggests that pattern recognition has a higher accuracy when performed by more experienced ultrasound practitioners, this study benefited from the involvement of level III gynaecological ultrasound practitioners for masses deemed malignant, unclassifiable, or where there was any form of uncertainty [27].

Limitations of this study include it being single-centred and conducted in a unit with experienced level II and level III gynaecological ultrasound practitioners which may not be the case in other settings. Our sample was heterogenous in that it included both symptomatic and asymptomatic women, however this does mimic clinical practice. Due to the absence of malignancy in this cohort, the performance of IOTA simple rules could not be assessed. Finally, because of the relatively small number of women requiring surgical intervention, the gold standard of histological diagnosis was only available for a few participants, meaning proxy markers of mass resolution/reduction in size had to be used. To overcome these limitations, a prospective multi-centre study is required. Ideally, this would examine unselected, asymptomatic women at fixed points such as 8-, 12- and 20-weeks’ gestation and 6 weeks and 3 months postpartum to give further time for spontaneous resolution.

The incidence rate in this study of 1% was comparable to the results of a large retrospective review [28]. Based on the Royal College of Obstetricians and Gynaecologists (RCOG) nomenclature for discussing risk, patients should be advised that adnexal masses in pregnancy are ‘uncommon’ [29]. The mean age at diagnosis in this study (32.02 years) mirrored that of two recent systematic reviews and also the mean age in the UK for women to have their first child [5,14,16].

The expectation in this cohort was that a large proportion of the masses detected would be physiological cysts that would self-resolve throughout the pregnancy [10]. This was shown to be the case, as the subjective impression was a ‘simple cyst’ in 76% of participants, and of these, 80% resolved. Additionally, when assessing all subjective impressions, in 74% of participants the mass had resolved by the time of the postpartum scan and there was a statistically significant decrease in the mean volume of masses over the course of the study. This information should assist in guiding clinicians when formulating follow-up plans and offering reassurance when counselling patients. The volume of the adnexal masses showed a statistically significant reduction throughout the study. However, it is important to note that the majority of these masses were simple cysts (76%), which are known to resolve spontaneously. Simple cysts accounted for most of the overall volume reduction, while other types of cysts, such as dermoid cysts, are less likely to resolve and tend to persist during pregnancy. Therefore, averaging the volume changes across different cyst types may obscure the distinct behaviours of each. For clarity, we have separated the data for simple and dermoid cysts in the analysis. The resolution rate for simple cysts was 80%, while only 28% of dermoid cysts resolved spontaneously. This distinction is important as it highlights the different natural courses of these cyst types during pregnancy and postpartum. The persistence of dermoid cysts should not be conflated with the spontaneous resolution of simple cysts

The distribution of subjective impressions mimicked that of large previous studies, with simple cysts being the most common, followed by dermoid cysts, endometriomas, and para-ovarian cysts [30]. The fact that a subjective impression was given for all masses in this study is likely a reflection of the scans being performed by experienced level II ultrasound practitioners, with support from level III practitioners.

Subjective impression of adnexal masses is based on pattern recognition and has a quoted accuracy of 92% in non-pregnant women in deciphering benign from malignant masses [31]. There is no data to date comparing this with pregnancy. In this study, histology was available in 14/274 participants. While this number is small, in 11/14 (79%) of cases, the subjective impression was correct. Additionally, the high-resolution rate of simple cysts and perseverance of endometriomas and dermoid cysts in this study suggest subjective impression is reliable during pregnancy.

While the change in subjective impression throughout the study was not statistically significant, it is of interest, as to date there appears to be no data regarding how commonly this occurs either when the same practitioner assesses a mass serially, or when different practitioners assess the same mass. Besides the aforementioned changes in the morphology of endometriomas, no other studies have assessed changes in other histological subtypes during pregnancy. In this study, the most common change was from either a simple to a haemorrhagic cyst or vice versa. This can be explained by spontaneous haemorrhage and resolution within a simple cyst and is commonly seen in clinical practice. A level III ultrasound practitioner was asked to review and confirm all these cases, as well as the two suspected BOT. As such, their input was deemed necessary in 18/274 (6.6%) of patients. In all cases, there was agreement except for one, where the diagnosis of a fibroma was made.

In this study, 99.3% of masses were classifiable based on IOTA simple rules which were higher when compared to studies in the non-pregnant cohort with quoted rates of 76% [21]. The reason for this may be multifactorial. Firstly, in this cohort, simple cysts, endometriomas, and dermoid cysts were most commonly diagnosed and have been shown to be largely classifiable, while malignancy and rarer benign tumours such as luteomas, which were not detected in this study, are more likely to be unclassifiable [21]. Secondly, all practitioners performing the ultrasounds had considerable experience both in gynaecological ultrasound and also in the use of IOTA simple rules, which has been shown to improve rates [27,31]. Thirdly, as most large studies assessing IOTA simple rules were conducted soon after its release in 2008, it would seem likely that with time, clinicians would have become more familiar with it, thus increasing the rates of classification. Supporting this is the fact that smaller, more recent studies of the IOTA simple rules found 89% to 93% of masses in non-pregnant women were classifiable [32,33]. While it was not a primary aim of this study, it was not possible to determine the sensitivity, positive likelihood ratio, negative likelihood ratio, positive predictive value, or accuracy of IOTA simple rules due to the absence of malignancy in our cohort.

As expected, the role of surgical intervention in this cohort was limited, with only two patients undergoing surgery in the antenatal period [7,10]. While one benefited from the detorsion of an ovary and subsequent uncomplicated pregnancy, the second underwent unnecessary intervention for a presumed BOT which was identified as a dermoid on histological examination. The pregnancy continued without complication. In the latter case, MRI findings were consistent with the ultrasound impression of a BOT. Since MRI was only utilized in one case, no conclusions can be drawn about its accuracy. To note, two recent systematic reviews failed to demonstrate the superiority of MRI over ultrasound but instead suggested a propensity for overdiagnosis of malignancy [14,16].

## 5. Conclusions

We suggest that in the absence of concerns regarding malignancy, torsion, or haemorrhagic rupture, there is no need for further monitoring of adnexal masses during the pregnancy, but rather a transvaginal ultrasound can be offered at least 6 weeks postpartum to give time for resolution and avoid unnecessary intervention. Further work is required to determine the benefit of screening the adnexa during routine antenatal ultrasound and the accuracy of ultrasound tools such as IOTA.

## Figures and Tables

**Figure 1 diagnostics-14-02182-f001:**
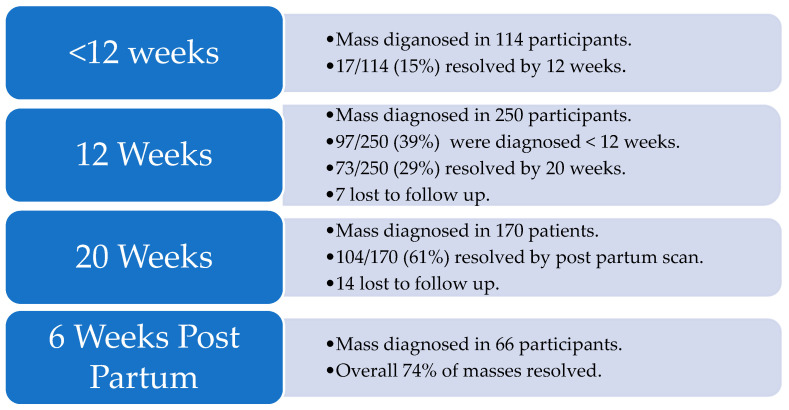
Masses detected, masses resolved, and loss to follow-up numbers at each study contact.

**Figure 2 diagnostics-14-02182-f002:**
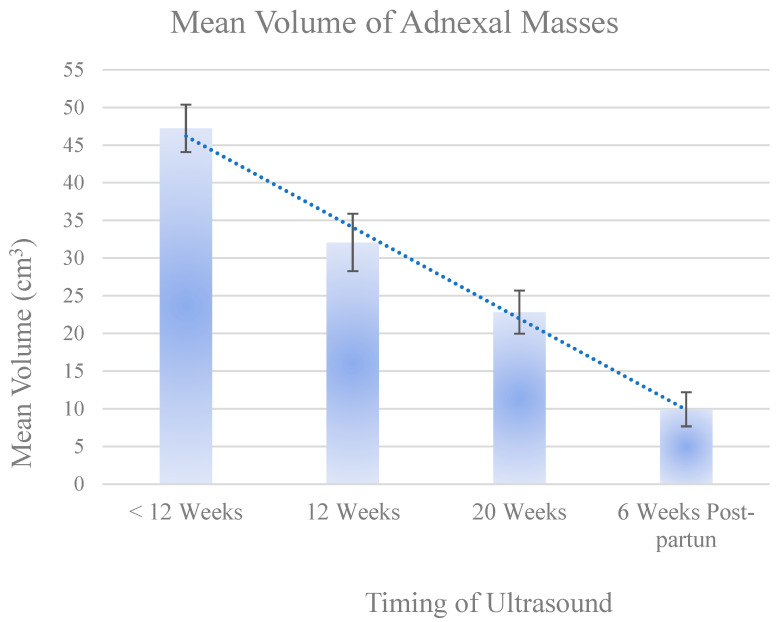
Mean volume change in adnexal masses over study time. The error bars represent the 95% confidence intervals (CI) for the mean volume of adnexal masses at each time point (prior to 12 weeks, at 12 weeks, 20 weeks, and 6 weeks postpartum). These were calculated using the standard error of the mean, which indicates the variability of the sample mean. The 95% CI offers a range within which we can be 95% confident that the true population mean lies. For the comparison between mean volumes prior to 12 weeks and at 12 weeks, the *p*-value was <0.0001. For the comparison between mean volumes prior to 12 weeks and at 20 weeks, the *p*-value was also <0.0001. Finally, for the comparison between prior to 12 weeks scans and 6 weeks postpartum, the *p*-value was 0.0026.

**Table 1 diagnostics-14-02182-t001:** Resolution rate during follow-up based on subjective impression.

Subjective Impression	N	Resolved	Persisted	% Resolved
Simple	208	167	41	80
Dermoid	25	7	18	28
Endometrioma	18	2	16	11
Hemorrhagic	9	4	5	44
Para ovarian	7	1	6	14
Torted Simple	2	2	0	100
Mucinous BOT *	2	0	2	0
Decidualized Endometrioma	1	0	1	0
Fibroma	1	0	1	0
Theca Luteal	1	1	0	100

* BOT = Borderline Ovarian Tumor.

**Table 2 diagnostics-14-02182-t002:** Histological diagnoses, timing of surgery, ultrasound and MRI impression.

Participant Number	Ultrasound Subjective Impression	MRI	Timing of Surgery	Histology
1	Mucinous BOT *	Mucinous BOT	Antenatal	Dermoid
2	Torted simple cyst	-	Antenatal	Serous cystadenoma (infarcted)
3	Dermoid	-	Caesarean	Dermoid
4	Dermoid	-	Caesarean	Dermoid
5	Endometrioma	-	Caesarean	Endometrioma
6	Dermoid	-	Caesarean	Dermoid
7	Dermoid	-	Caesarean	Dermoid
8	Dermoid	-	Caesarean	Dermoid
9	Mucinous BOT	-	Postpartum	Dermoid
10	Simple	-	Postpartum	Mucinous cystadenoma
11	Endometrioma	-	Postpartum	Endometrioma
12	Hydrosalpinx	-	Postpartum	Dermoid
13	Simple	-	Postpartum	Serous cystadenoma
14	Dermoid	-	Postpartum	Dermoid

* BOT = Borderline Ovarian Tumor.

## Data Availability

Data can be obtained by contacting the corresponding author.

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
