# Peer review of "Adnexal Masses in Pregnancy: A Single-Centre Prospective Observational Cohort Study"

_diagnostics, 2024, doi:10.3390/diagnostics14192182_

Round 1

Reviewer 1 Report

Comments and Suggestions for Authors

This paper is a study of the follow-up of ovarian tumors during pregnancy by ultrasonography using IOTA.

Although this is a single-arm study, it is a prospective study of a very large number of cases, and as a result, it is concluded that ovarian tumors during pregnancy should be examined by ultrasound, and if there are no findings suspicious of malignancy or borderline malignancy, the natural course of the disease is acceptable.

Although these results are acceptable for publication because they are helpful for ovarian tumors during pregnancy, we have the following doubts that should be resolved.

Why do the authors use the protocol of taking echoes at discovery, 12 weeks, 20 weeks, and 6 weeks postpartum?

The authors mention that the volume of cysts shrinks, but most of them are simple cysts, so is it appropriate to discuss them in the same place as other dermoid cysts as if they were averages? Doubts arise.

Although this was not the case in the present cases, there are many reports of dermoid cysts that have torsion or rupture during pregnancy, requiring emergency surgery. Since dermoid cysts do not disappear, it would be better to avoid saying that surgery is not necessary because the cyst disappears.

Author Response

Comment 1: Why do the authors use the protocol of taking echoes at discovery, 12 weeks, 20 weeks, and 6 weeks postpartum?

Response: Many thanks for your comment. 

The study uses a protocol of taking ultrasound scans at the time of detection, 12 weeks, 20 weeks, and 6 weeks postpartum to monitor the progression or resolution of adnexal masses during pregnancy and after delivery. This timeline aligns with key milestones in pregnancy:

  1. Detection and Early Pregnancy (up to 12 weeks): The early pregnancy scan at 12 weeks allows for the identification and characterization of any adnexal masses that may have appeared at the start of the pregnancy.

  2. Anomaly Scan at 20 weeks: By this time, any persisting adnexal masses can be re-evaluated, and decisions regarding the need for further follow-up or intervention can be made. This is also when the uterus has expanded, and the adnexal structures can be visualized trans-abdominally.

  3. Postpartum (6 weeks): At 6 weeks postpartum, the masses can be assessed for resolution, as the majority of masses detected during pregnancy tend to resolve spontaneously. This timeframe also allows for the possibility of surgical intervention if the mass has not resolved or if complications arose during pregnancy.

By following this protocol, the study aimed to capture the natural history of adnexal masses over the course of pregnancy and into the postpartum period, providing insight into when these masses resolve or if they require further management

We have highlighted this in our methods: Page 2, Line 75-81. "The timing of ultrasound scans was chosen to correspond with critical points in pregnancy management: detection or routine dating scan (around 12 weeks) allows for early identification of adnexal masses; the anomaly scan (18-22 weeks) enables further assessment as the pregnancy progresses and the uterus expands, which may impact visibility and mass characteristics; and the postpartum scan at 6 weeks is included to assess the natural resolution of masses after pregnancy when most adnexal masses tend to resolve spontaneously"

Comment 2: The authors mention that the volume of cysts shrinks, but most of them are simple cysts, so is it appropriate to discuss them in the same place as other dermoid cysts as if they were averages? Doubts arise.

Response: The reviewer raises a valid concern regarding the interpretation of cyst volume reduction. Since most of the cysts in the study were simple cysts, and these are expected to resolve spontaneously during pregnancy, it may not be entirely appropriate to discuss volume changes across all cyst types as a single averaged result without distinguishing between cyst types, especially when comparing simple cysts to more complex types like dermoids. 

We have added the following to our discussion (Page 10 , lines 324-334) "

The volume of the adnexal masses showed a statistically significant reduction over the course of the study. However, it is important to note that the majority of these masses were simple cysts (76%), which are known to resolve spontaneously. Simple cysts accounted for most of the overall volume reduction, while other types of cysts, such as dermoids, are less likely to resolve and tend to persist during pregnancy. Therefore, averaging the volume changes across different cyst types may obscure the distinct behaviors of each. To provide more clarity, we have separated the data for simple cysts and dermoid cysts in the analysis. The resolution rate for simple cysts was 80%, while only 28% of dermoid cysts resolved spontaneously. This distinction is important as it highlights the different natural courses of these cyst types during pregnancy and postpartum. The persistence of dermoid cysts should not be conflated with the spontaneous resolution of simple cysts​"

Comment 3: Although this was not the case in the present cases, there are many reports of dermoid cysts that have torsion or rupture during pregnancy, requiring emergency surgery. Since dermoid cysts do not disappear, it would be better to avoid saying that surgery is not necessary because the cyst disappears.

Response: We highly appreciate this point and now we have added to our discussion Page 10, line 330 "The resolution rate for simple cysts was 80%, while only 28% of dermoid cysts resolved spontaneously. This distinction is important as it highlights the different natural courses of these cyst types during pregnancy and postpartum. The persistence of dermoid cysts should not be conflated with the spontaneous resolution of simple cysts"

Reviewer 2 Report

Comments and Suggestions for Authors

The authors conducted the prospective study to reveal the nature of adnexal masses diagnosed during pregnancy. It is interesting and worthwhile as a paper.

There are some comments about this article to publish in the journal.

・According to table 1, 28% of dermoid cysts were resolved spontaneously. However, teratomas are not expected to resolve spontaneously and needed to have surgical treatment in many cases. This point should be discussed.

・Are the incidence rate and 95 %CI described on line 121 correct?

・The authors should explain the error bar in figure 2.

・“BOT” is not defined in the text. The authors should use the abbreviations in accordance with the author’s guideline.

・One patient had the andexal mass with a malignant feature classified with IOTA criteria. However, the clinical course of the patient was unclear. The authors should provide the information in more detail.

Author Response

Comment 1: According to table 1, 28% of dermoid cysts were resolved spontaneously. However, teratomas are not expected to resolve spontaneously and needed to have surgical treatment in many cases. This point should be discussed.

Response: Many thanks for your comment. We have now included in discussion page 10, line 324-334 "

The volume of the adnexal masses showed a statistically significant reduction over the course of the study. However, it is important to note that the majority of these masses were simple cysts (76%), which are known to resolve spontaneously. Simple cysts accounted for most of the overall volume reduction, while other types of cysts, such as dermoids, are less likely to resolve and tend to persist during pregnancy. Therefore, averaging the volume changes across different cyst types may obscure the distinct behaviors of each. To provide more clarity, we have separated the data for simple cysts and dermoid cysts in the analysis. The resolution rate for simple cysts was 80%, while only 28% of dermoid cysts resolved spontaneously. This distinction is important as it highlights the different natural courses of these cyst types during pregnancy and postpartum. The persistence of dermoid cysts should not be conflated with the spontaneous resolution of simple cysts​"

Comment 2: Are the incidence rate and 95 %CI described on line 121 correct?

Response: 2.6% is correct, but we deleted 0.26 as typo

Comment 3: The authors should explain the error bar in figure 2.

Response: In this case, the error bars help to illustrate the degree of certainty regarding the changes in mass volume over time. Narrower error bars indicate less variability and higher confidence in the observed mean volume, while wider error bars suggest greater variability. These intervals are particularly important when comparing volume changes across different time points, as they allow for the assessment of whether observed changes are statistically significant or could be attributed to sampling variation

We have now added to the image footage "

The error bars represent the 95% confidence intervals (CI) for the mean volume of adnexal masses at each time point (prior to 12 weeks, at 12 weeks, 20 weeks, and 6 weeks postpartum). These were calculated using the standard error of the mean , which provides an indication of the variability of the sample mean. The 95% CI offers a range within which we can be 95% confident that the true population mean lies.

Comment 4: BOT” is not defined in the text. The authors should use the abbreviations in accordance with the author’s guideline.

Response: In line 71 when first mentioned BOT, we have added the definition Borderline Tumor. We have now included definition  of BOT in the text in all places. Highlighted Page 3, line 113. Page 5, line 175. Page 6, line 196. Page 6, line 202. Page 6, line 211. Page 10, line 368

Comment 5: One patient had the andexal mass with a malignant feature classified with IOTA criteria. However, the clinical course of the patient was unclear. The authors should provide the information in more detail.

Response: We have added in page 8, line 281: The latter ovarian cyst was a multilocular cyst with increased vascularity (IOTA color score of 4). The patient had the ovarian cyst removed after delivery and it was a mature cystic teratoma.

"